# Learning Pruning-Friendly Networks via Frank-Wolfe: One-Shot, Any-Sparsity, And No Retraining

**Miao Lu[1*], Xiaolong Luo[1*], Tianlong Chen[2], Wuyang Chen[2], Dong Liu[1], Zhangyang Wang[2]**
[1]University of Science and Technology of China, [2]University of Texas at Austin
{lumiao,lxl213}@mail.ustc.edu.cn, dongeliu@ustc.edu.cn
{tianlong.chen,wuyang.chen,atlaswang}@utexas.edu

## Abstract

We present a novel framework to train a large deep neural network (DNN) for only *once*, which can then be pruned to *any sparsity ratio* to preserve competitive accuracy *without any re-training*. Conventional methods often require (iterative) pruning followed by re-training, which not only incurs large overhead beyond the original DNN training but also can be sensitive to retraining hyperparameters. Our core idea is to re-cast the DNN training as an explicit *pruning-aware* process: that is formulated with an auxiliary $K$-sparse polytope constraint, to encourage network weights to lie in a convex hull spanned by $K$-sparse vectors, potentially resulting in more sparse weight matrices. We then leverage a stochastic Frank-Wolfe (SFW) algorithm to solve this new constrained optimization, which naturally leads to sparse weight updates each time. We further note an overlooked fact that existing DNN initializations were derived to enhance SGD training (e.g., avoid gradient explosion or collapse), but was unaligned with the challenges of training with SFW. We hence also present the first learning-based initialization scheme specifically for boosting SFW-based DNN training. Experiments on CIFAR-10 and Tiny-ImageNet datasets demonstrate that our new framework named *SFW-pruning* consistently achieves the state-of-the-art performance on various benchmark DNNs over a wide range of pruning ratios. Moreover, SFW-pruning only needs to train once on the same model and dataset, for obtaining arbitrary ratios, while requiring neither iterative pruning nor retraining. Codes are available in https://github.com/VITA-Group/SFW-Once-for-All-Pruning.

## 1 Introduction

Deep neural networks (DNNs) achieve tremendous empirical success in various machine learning applications, but this usually requires a huge model size and high computational cost, challenging the usage of such models in various real-time and multi-platform applications. For example (Cai et al., 2019), mobile applications on App stores have to support a diverse range of hardware devices, from high-end flagships to low-end ones, which need different model capacities for different scenarios. As a remedy, neural network compression (Han et al., 2015) tries to slim various DNNs to improve both the computation and memory efficiency. One important way of network compression is network pruning, which prunes individual weights (unstructured pruning, Han et al. (2015)) or channels (structured pruning, Li et al. (2016)) of the network, causing little degeneration in the test performance. Usually, modern DNN pruning techniques require retraining or fine-tuning of the compressed network, either in a one-shot manner or iterative manner (Frankle & Carbin, 2018). When one aims to deploy the compressed neural network in different platforms, retraining the pruned model requires huge computational cost, resulting in excessive energy consumption.

To tackle such an efficiency problem, in this paper, we aim to answer the question of whether we can design an efficient pruning method that does not need retrain the neural network. Specifically, we hope to design *one-shot* unstructured pruning algorithms, which can guarantee *consistent* and *competitive* model performance under *varying* pruning ratios *without retraining* the neural network. One-shot pruning is a natural choice since retraining is not allowed. Intuitively, given a sparsity

ratio, one can leave the most important individual weights (weight-magnitude unstructured pruning (Han et al., 2015)) untouched and the less important ones pruned according to their relative "absolute values". In practice, different pruning ratio requirements correspond to different computational and memorial budgets. However, previous pruning methods cannot address our question, since they mainly find those important weights via standard gradient-based optimization methods during training, e.g., stochastic gradient descent (SGD), whose performance will degrade much if retraining is prohibited, especially at large pruning ratios (See Figure 1(a)). The reason is that such optimization methods fail to take the desired sparse structures in the pruned model into consideration. The process of finding those most important weights should be integrated with the optimization method.

In view of those, we innovate to re-cast DNN training as an explicit *pruning-aware* process, which is formulated with an auxiliary $K$-sparse polytope constraint to encourage the DNN weights to lie in the convex hull spanned by $K$-sparse vectors. Correspondingly, we also propose to train the DNN via solving the new constrained optimization problem using the stochastic Frank-Wolfe (SFW) algorithm (Reddi et al., 2016; Hazan & Luo, 2016) that can more organically handle constraints than SGD-type algorithms. SFW results in sparse weight updates and continually pushes less important weights to smaller magnitudes. We observe DNNs trained in this way to have more small-value (*yet non-zero*) weights that can be gradually and smoothly removed when increasing the (one-shot) pruning ratio. Applying one-shot weight magnitude pruning then yields consistently strong test performance across different pruning ratios *without* further retraining or fine-tuning.

Moreover, we propose a new initialization scheme for SFW-based training. We note that SFW algorithm is *not* a gradient-based optimization method, and the widely used initialization schemes like Kaiming initialization (He et al., 2015) for gradient-based methods do not fit SFW algorithm since the latter does not use the gradient to update weights. Our proposed initialization scheme is organically designed for SFW algorithm and is formulated as a meta learning problem, drawing motivations from Zhu et al. (2021). It learns the layer-wise initialization scaling factors that lead to the largest loss reduction in the first SFW training step. We demonstrate that with the new initialization scheme, our proposed one-shot pruning algorithm can consistently achieve better test performance under different pruning ratios without retraining. Now we summarize our main contributions.

- We explicitly re-cast DNN training as a pruning-aware process formulated under an auxiliary $K$-sparse polytope constraint, based on which we propose a new SFW-pruning framework that trains a DNN via solving a new constraint optimization problem using SFW and prunes the DNN in a one-shot fashion with no retraining.

- We customize a meta-learning-based initialization scheme for SFW-based DNN training, which leads to more consistent and competitive performance under varying pruning ratios.

- We empirically demonstrate that our proposed SFW-pruning framework is applicable across different architectures and datasets, achieving the state-of-the-art performance consistently over a wide range of pruning ratios without retraining.

## 2 RELATED WORK

**Neural Network Pruning and Efficient Deployment.** With larger, deeper, and more sophisticated models, DNNs achieved incredible success over the last decade. However, large models introduce high computation and memory costs, making them difficult in actual application and deployment. Various methods, including knowledge distillation (Hinton et al., 2015; Romero et al., 2014), low-rank factorization (Denton et al., 2014; Yu et al., 2017), quantization (Courbariaux et al., 2016; Rastegari et al., 2016; Wu et al., 2018), and pruning (Han et al., 2015; Li et al., 2016; Liu et al., 2018), have been proposed to deploy large models in resource-constrained devices efficiently. Among these methods, pruning has become a research hotspot for its ability to maintain high performance.

Typically, pruning can be roughly divided into two main branches: unstructured (Han et al., 2015) and structured (Hu et al., 2016). Unstructured pruning is to remove individual parameters in the networks. A common example, Iterative Magnitude Pruning (IMP) (Han et al., 2015; Frankle & Carbin, 2018; Chen et al., 2020b;a), repeats training and pruning cycles to attain a sparse network. Although IMP works well, its high computation cost motivates more efficient methods (Lee et al., 2018; Wang et al., 2020a; Tanaka et al., 2020). Another major branch is structured pruning. It

considers parameters in groups, removing entire neurons, filters, or channels (He et al., 2017; Li et al., 2016; Guo et al., 2021) which can more directly save energy and memory (Luo & Wu, 2020).

Practical devices often vary in their on-board resource availability (Cai et al., 2019; Wang et al., 2020b). To tackle this problem, pruning the network to different sizes and retraining is one naive way, yet costly. (Han et al., 2015; He et al., 2017; Li et al., 2016). Another approach is to design efficient and scalable neural network architecture such as MobileNet (Howard et al., 2017) and ShuffleNets (Ma et al., 2018; Zhang et al., 2018). However, these methods are computationally expensive or need human-based design. To this end, Cai et al. (2019) pioneer on a "once for all" scheme that can flexibly obtain a large number of sub-networks to meet different resource constraints from a pre-trained "super network", but their pre-training cost is gigantic and re-training is needed for each sub-network for restoring the optimal performance. In our work, we view efficient deployment from a novel aspect, i.e. training a neural network to naturally possess scalable weight sparsity, while keeping the training cost comparable with normal one-pass training. Then with only a single-shot pruning, we can get the desired sub-network with any sparsity without retraining.

**Stochastic Frank-Wolfe.** Frank–Wolfe (FW) (Frank et al., 1956) is a classical non-gradient-based method for convex optimization. In recent years, FW has been applied in stochastic non-convex optimization (Reddi et al., 2016; Hazan & Luo, 2016), named as stochastic Frank–Wolfe (SFW) algorithm. Hazan & Luo (2016) perform a theoretical analysis of the standard SFW algorithm with a convergence rate $\mathcal{O}(1/t)$ with $\Theta\left(t^2\right)$ growing batch-sizes. Several works further extend SFW-based algorithms in the deep learning field and propose various variants of SFW (Yurtsever et al., 2019; Shen et al., 2019; Xie et al., 2019; Zhang et al., 2020; Combettes et al., 2020). Momentum is found to be a crucial part of SFW in training neural networks (Cutkosky & Orabona, 2019; Mokhtari et al., 2020; Chen et al., 2018). Recently, (Pokutta et al., 2020) apply a momentum based SFW algorithm in training NN and achieve high accuracy. Inspired from those prior arts, we propose SFW-based NN training algorithm for one-shot pruning without retraining. We additionally investigate the previously overlooked problem of SFW initialization, and present a meta learning-based scheme.

**Neural Networks Initialization.** An inappropriate initialization can lead to deep networks weights and gradients exploding or vanishing, challenging training deep neural networks. Several standard initialization (Glorot & Bengio, 2010; He et al., 2015) methods are designed for gradient-based optimization methods to keep the variance per layer balanced. Further, Glorot & Bengio (2010) and He et al. (2015)'s assumptions no longer hold for more complex architectures, motivating newer methods (Dauphin & Schoenholz, 2019; Zhang et al., 2019; Bachlechner et al., 2020). However, most of these are tied to enhancing SGD-based training. Since the $K$-sparse SFW algorithm is non-gradient based, new initialization methods need to be customized.

# 3 STOCHASTIC FRANKE-WOLFE PRUNING FRAMEWORK

In this section, we formulate the DNN training process as an explicit pruning-aware process with an auxiliary $K$-sparse polytope constraint, and we solve the corresponding constrained optimization problem via a stochastic Franke-Wolfe (SFW) algorithm. Specifically, we formulate the pruning-aware process and review the basics of SFW algorithm in Section 3.1. In Section 3.2, we present our proposed *SFW-pruning framework* based on the pruning-aware process and one-shot pruning, where we consider weight-magnitude unstructured pruning.

## 3.1 PRUNING-AWARE DNN TRAINING AND STOCHASTIC FRANK-WOLFE ALGORITHM

Our core idea is to re-cast the DNN training process as an explicit pruning-aware process, which is achieved via an auxiliary $K$-sparse polytope constraint. Formally, given a dataset $\mathbb{D} = \{(\mathbf{x}_i, y_i)\}_{i=1}^n$ where $(\mathbf{x}_i, y_i) \in \mathcal{X} \times \mathcal{Y}$ and a loss function $\ell : \mathcal{Y} \times \mathcal{Y} \mapsto \mathbb{R}_+$, e.g., cross-entropy loss, we aim to train a deep neural network $f(\boldsymbol{\theta}; \cdot) : \mathcal{X} \mapsto \mathcal{Y}$ that minimizes the following *pruning-aware objective*:

$$\min_{\boldsymbol{\theta} \in \mathcal{C}} \frac{1}{n} \sum_{i=1}^n \ell(f(\boldsymbol{\theta}; \mathbf{x}_i), y_i) := \min_{\boldsymbol{\theta} \in \mathcal{C}} L(\boldsymbol{\theta}) \qquad (1)$$

To make the training process pruning-aware, we restrict the feasible parameter $\boldsymbol{\theta}$ in a convex region $\mathcal{C}$, which potentially results in more sparse weight matrices and will be more pruning-friendly for

only a small percentage of the weights are of large magnitudes. Specifically, we choose $\mathcal{C}$ as a $K$-sparse polytope, and we solve (1) efficiently via a stochastic Frank–Wolfe (SFW) algorithm. We review the $K$-sparse polytope and the basics of SFW below and further explain our motivations.

**Stochastic Frank-Wolfe and $K$-Sparse Polytope Constraints.** SFW is a simple projection-free first-order algorithm for solving convex constraint optimization problems (Reddi et al., 2016; Hazan & Luo, 2016; Yurtsever et al., 2019; Shen et al., 2019; Xie et al., 2019; Zhang et al., 2020; Combettes et al., 2020). Consider the constrained optimization objective (1) and denote the neural network weights learned by SFW by $\boldsymbol{\theta}_t$. At each iteration $t$, SFW first calls a *linear minimization oracle* $\boldsymbol{v}_t = \text{LMO}_{\mathcal{C}}(\widehat{\nabla}_{\boldsymbol{\theta}} L(\boldsymbol{\theta}_t)) = \arg\min_{\boldsymbol{v} \in \mathcal{C}} \langle \widehat{\nabla}_{\boldsymbol{\theta}} L(\boldsymbol{\theta}_t), \boldsymbol{v} \rangle$, where $\widehat{\nabla}_{\boldsymbol{\theta}} L(\boldsymbol{\theta}_t)$ estimates the full gradient, e.g., gradient on a minibatch. Given $\boldsymbol{v}_t$, SFW updates $\boldsymbol{\theta}_t$ along the direction of $\boldsymbol{v}_t$ by a convex combination $\boldsymbol{\theta}_{t+1} = \boldsymbol{\theta}_t + \alpha_t(\boldsymbol{v}_t - \boldsymbol{\theta}_t) = \alpha_t \boldsymbol{v}_t + (1 - \alpha_t)\boldsymbol{\theta}_t$ with the learning rate $\alpha_t \in [0, 1]$. This keeps $\boldsymbol{\theta}_t$ always in $\mathcal{C}$ and saves any projection step. In neural network training, to further improve the training performance and test accuracy, momentum is also introduced into SFW algorithm (Xie et al., 2019; Pokutta et al., 2020). We conclude the corresponding SFW algorithm in Algorithm 1.

A $K$-sparse polytope in $\mathbb{R}^p$ of radius $\tau > 0$, $p \in \mathbb{N}$, denoted by $\mathcal{C}(K, \tau)$, is the intersection of the $L^1$-ball $\mathcal{B}_1(\tau K)$ and the $L^\infty$-ball $\mathcal{B}_\infty(\tau)$ of $\mathbb{R}^p$. One can obtain $\mathcal{C}(K, \tau)$ by spanning all the vectors in $\mathbb{R}^p$, which have exactly $K$ non-zero coordinates and the absolute value of the non-zero entries are $\tau$. It holds that (Pokutta et al., 2020) for any $\boldsymbol{m} \in \mathbb{R}^p$ the oracle $\boldsymbol{v} = \text{LMO}_{\mathcal{C}(K,\tau)}(\boldsymbol{m})$ is given by

$$(\boldsymbol{v})_i = \begin{cases} -\tau \cdot \text{sign}((\boldsymbol{m})_i) & \text{if } (\boldsymbol{m})_i \text{ is in the largest } K \text{ coordinates of } \boldsymbol{m}, \\ 0 & \text{otherwise,} \end{cases} \quad \forall 1 \leq i \leq p, \quad (2)$$

which is a vector with exactly $K$ non-zero entries. When applying the $K$-sparse polytope constraint in deep neural network training objective (1), we add the constraint on each layer of the network. In other words, if we write out $\boldsymbol{\theta}$ as $\boldsymbol{\theta} = (\boldsymbol{W}_1, \boldsymbol{b}_1, \cdots, \boldsymbol{W}_L, \boldsymbol{b}_L)$ for layer-wise weight parameter $\boldsymbol{W}_l$ and bias parameter $\boldsymbol{b}_l$, then each of the parameters is paired with a $K$-sparse polytope constraint $\mathcal{C}(K_l, \tau)$. Here $K_l$ may vary between the different layers. For notation simplicity, in the sequel, we denote such a layer-wise constraint as $\boldsymbol{\theta} \in \mathcal{C}(\{K_l\}_{l=1}^L, \tau)$ without making any confusion.

**Why $K$-Sparse Polytope Constraints?** In each step of our optimization, the linear minimization oracle of a $K$-sparse polytope constraint returns a update vector with exactly $K$ non-zero coordinates, which is then weighted-averaged with the current $\boldsymbol{\theta}_t$ according to the SFW algorithm (accumulated from the "ensemble" of $K$ non-zero coordinate vectors, from all past update steps). In other words, training with a $K$-sparse constraints amounts to a "voting" process, by all step updates, on which $K$ elements "should be non-zero", and the resulting weights could be viewed as the soft voting consensus. By adding such a training constraint, each SFW step pushes less important weights smaller since they are averaged with zero, and those important weights are enhanced meanwhile.

As Figure 1(a) shows, compared to DNNs trained by SGD, those trained by SFW with $K$-sparse polytope constraints appear to have much more smaller weights (*but not exactly zero*), and less large ones. Moreover, the amounts of weights, at different magnitude levels, change more "smoothly and "continually" in SFW-trained weights compared to that of SGD. With such a weight distribution, when the pruning ratio increases, there will be no sudden "jump" to removing small values to impacting larger values) when the pruning ratio increases, yielding competitive test accuracies across the whole spectrum of pruning ratios, even without needing retraining. We refer to Appendix B.1 for more computational details about $K$-Sparse Polytope Constraints.

### 3.2 STOCHASTIC FRANK–WOLFE ONE-SHOT PRUNING WITHOUT RETRAINING

Previously, we have motivated the usage of $K$-sparse polytope constraint in the pruning-aware DNN training objective (1) and using SFW algorithm in the training process. In this subsection, we first propose our *SFW-pruning* framework for deep neural network one-shot pruning *without retraining* based on SFW neural network training with $K$-sparse polytope constraints. After, we provide some fine-grained analysis of the hyperparameters involved in the newly proposed pruning framework.

**Stochastic Frank–Wolfe Pruning Framework.** We now present our one-shot pruning framework, where a DNN is trained using the pruning-aware objective (1) with $K$-sparse constraints via SFW *for only once* and then undergoes a one-shot pruning (Algorithm 2). The SFW training phase (Line 5 to 7) is pruning-aware, which aims to find a proper DNN weight $\boldsymbol{\theta}_T$ that can minimize the training

---

**Algorithm 1:** Stochastic Frank-Wolfe with Momentum for Deep Neural Network Training

---

1: **Input:** Dataset $\mathbb{D} = \{(\mathbf{x}_i, y_i)\}_{i=1}^n$, deep neural network $f(\boldsymbol{\theta}; \cdot)$, convex constraint region $\mathcal{C}$, initialization point $\boldsymbol{\theta}_0 \in \mathcal{C}$, linear minimization oracle $\text{LMO}_\mathcal{C}$, number of steps $T$, initial learning rate $\alpha_0 \in [0, 1]$, learning rate scheme $\text{lr\_scheme}$, momentum $\rho \in [0, 1]$.
2: **Output:** final point $\boldsymbol{\theta}_T = \text{SFW}(\mathbb{D}, f, \mathcal{C}, \text{LMO}_\mathcal{C}, \boldsymbol{\theta}_0, T, \alpha_0, \text{lr\_scheme}, \rho)$.
3: **for** $t = 1, \cdots, T$ **do**
4:     Update learning rate $\alpha_t \leftarrow \text{lr\_scheme}(t)$ and gradient estimator $\widehat{\nabla}_{\boldsymbol{\theta}} L(\boldsymbol{\theta}_{t-1})$.
5:     Update momentum vector $\boldsymbol{m}_t \leftarrow (1 - \rho)\boldsymbol{m}_{t-1} + \rho\widehat{\nabla}_{\boldsymbol{\theta}} L(\boldsymbol{\theta}_{t-1})$.
6:     Solve linear minimization oracle $\boldsymbol{v}_t \leftarrow \text{LMO}_\mathcal{C}(\boldsymbol{m}_t)$.
7:     Update neural network weights $\boldsymbol{\theta}_t \leftarrow \boldsymbol{\theta}_{t-1} + \alpha_t(\boldsymbol{v}_t - \boldsymbol{\theta}_t)$.
8: **end for**

---

**Algorithm 2:** Stochastic Frank-Wolfe Pruning Framework (SFW-Pruning)

---

1: **Input:** Dataset $\mathbb{D}$, deep neural network $f(\boldsymbol{\theta}_0; \cdot)$, diameter $\tau$, sparsity $\{K_l\}_{l=1}^L$, initial weight $\boldsymbol{\theta}_0 \in \mathcal{C}(\{K_l\}_{l=1}^L, \tau)$, linear minimization oracle $\text{LMO}_{\mathcal{C}(\{K_l\}_{l=1}^L, \tau)}$, training epoch $T$, initial learning rate $\alpha_0 \in [0, 1]$, learning rate scheme $\text{lr\_scheme}$, momentum $\rho \in [0, 1]$, desired pruning ratio $s$, $\text{Initialization\_Scheme}$ (bool), and $\text{Pruning\_Procedure}$ (weight-magnitude unstructured pruning).
2: **Output:** a pruned neural network $f(\boldsymbol{\theta}_T^s; \cdot)$ with pruning sparsity ratio $s$.
3: # Sparse training phase using SFW with $K$-sparse constraints
4: **if** $\text{Initialization\_Scheme}$ is True **then**
5:     $\boldsymbol{\theta}_0 \leftarrow \text{SFWInit}(f(\boldsymbol{\theta}_0; \cdot), \boldsymbol{\theta}_0)$.
6: **end if**
7: Train the deep neural network $f$ on $\mathbb{D}$ via SFW, i.e., set
    $\boldsymbol{\theta}_T \leftarrow \text{SFW}(\mathbb{D}, f, \mathcal{C}(\{K\}_{l=1}^L, \tau), \text{LMO}_{\mathcal{C}(\{K\}_{l=1}^L, \tau)}, \boldsymbol{\theta}_0, T, \alpha_0, \text{lr\_scheme}, \rho)$.
8: # One-shot magnitude pruning phase
9: Pruning the deep neural network $f(\boldsymbol{\theta}_T; \cdot)$ via $\text{Pruning\_Procedure}$ and get $f(\boldsymbol{\theta}_T^s; \cdot)$.

---

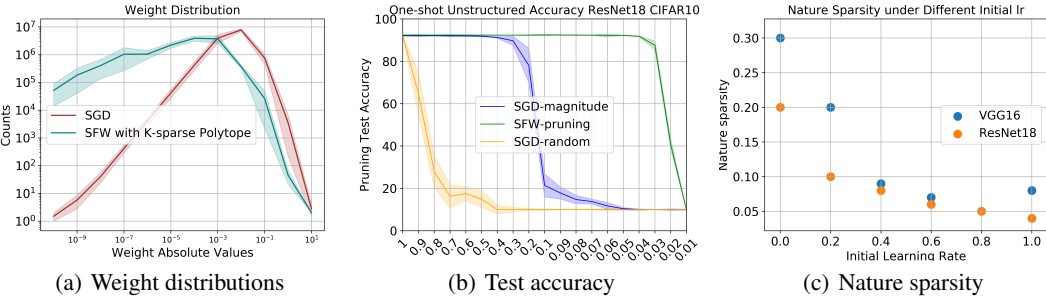

(a) Weight distributions  (b) Test accuracy  (c) Nature sparsity

Figure 1: (a) shows the weight magnitude distributions of DNNs trained by SGD and SFW respectively, averaged over 2 architectures {ResNet18, VGG16} and 2 datasets {CIFAR-10, CIFAR-100}. See Appendix B.3 for weight distributions for each combination. (b) shows the test accuracy under three different settings: SFW-pruning framework; SGD training and one-shot magnitude pruning; and SGD training and random pruning. The $x$-axis is the desired pruning ratio. (c) shows the nature sparsity of two architectures {ResNet18, VGG16} trained by SFW with $K$-sparse polytope constraints on CIFAR-10 dataset with $\alpha_0 \in \{0.1, 0.2, 0.4, 0.6, 0.8, 1.0\}$.

loss while being pruning-friendly. The following pruning procedure (Line 8 to 9) is adpoted with standard weight-magnitude unstructured pruning (Han et al., 2015; Frankle & Carbin, 2018) (prunes individual weights in the DNN). For practical use, one can choose either of the pruning procedures. We refer to technical details of the two pruning procedures to experiments in Section 5. In Figure 1(b), we demonstrate the effectiveness of our framework using ResNet18 (He et al., 2016) on the CIFAR-10 (Krizhevsky et al., 2009) dataset for unstructured pruning.

Despite the simplicity, the high-level idea of Algorithm 2 is intuitive: the discovery of sparse structures and pruning-aware DNN weights should be integrated with the *optimization process*, especially when retraining is prohibited. We further demonstrate the effectiveness of Algorithm 2 via extensive experiments on different datasets and DNN architectures with unstructured pruning, achieving competitive test performance consistently across various pruning ratios, which is not achievable by random pruning or non-pruning-aware optimization methods.

**Fine-grained Analysis of SFW Sparse Training.** Proper choices of several hyperparameters of Algorithm 2 play an important in discovering pruning-friendly weights. We note that intuitively it is a trade-off between the test performance and the one-shot pruning ratio of the DNN. But given the same pruning ratio, our framework can find more pruning-friend weights than random pruning or other optimization methods without sparsity awareness. We refer to Appendix B.1 for a detailed ablation study of the hyperparameters in Algorithm 2.

$K$-*sparse constraint parameters $\{K_l\}_{l=1}^L$ and $\tau$:* $\{K_l\}_{l=1}^L$ controls the sparsity of the linear minimization oracle solution, and $\tau$ controls the magnitude, which together influence the distribution of the learned parameters. Experimental results show that relatively smaller choices of these hyperparameters can keep the test performance undamaged under higher one-shot pruning ratios, at the cost of relatively lower full model test performance. Empirically, we find $K = 5\%$ and $\tau = 15$[1] to be our default good choices that transfer well across DNNs and datasets.

*Learning rate $\alpha$:* Recall that at step $t$, we update $\boldsymbol{\theta}_t$ by $\boldsymbol{\theta}_{t+1} = \boldsymbol{\theta}_t + \alpha_t(\boldsymbol{v}_t - \boldsymbol{\theta}_t) = \alpha_t \boldsymbol{v}_t + (1 - \alpha_t)\boldsymbol{\theta}_t$ which is the a convex combination of $\boldsymbol{v}_t$, the linear minimization orcale solution, and $\boldsymbol{\theta}_t$, the current parameter. Note that $\boldsymbol{v}_t$ is $K$-sparse, which means that a larger learning rate (i.e., $\alpha_t \boldsymbol{v}_t$ has larger weight in the convex combination) may result in sparser parameter $\boldsymbol{\theta}_{t+1}$. This observation indicates that one can (dynamically) control the nature sparsity[2] of the neural network by controlling the scale of the learning rate $\alpha_t$. We refer to Figure 1(c) for a demonstration of this phenomenon.

# 4 INITIALIZATION SCHEME FOR STOCHASTIC FRANK-WOLFE ALGORITHM

Most widely used initialization schemes for gradient-based methods, like Kaiming initialization (He et al., 2015), ensure that the mean of activations is zero, and the variance of the activations stays the same across layers to avoid gradient explosion or collapse. However, the SFW algorithm does not use the gradient to update the weights, making such initialization schemes unaligned with the challenges of training with SFW. A new initialization designed for SFW is desired.

To this end, we introduce a new initialization scheme by adapting the idea from Zhu et al. (2021). The goal is to learn the best initialization $\boldsymbol{\theta}_0$ in the sense that it can allow the maximal loss reduction in the first SFW step, which we hope can further bring more competitive and consistent pruning test performance in our SFW-pruning paradigm. Specifically, the proposed method initializes each layer of the neural network with a uniform distribution, which is then multiplied by a layer-wise scaling parameter learned by the algorithm, resulting in a maximal one-SFW-step loss reduction.

**Methodology.** We first initialize the parameter $\boldsymbol{\theta}_0$ of weight matrices $\{\boldsymbol{W}_l\}_{l=1}^L$ and bias vectors $\{\boldsymbol{b}_l\}_{l=1}^L$ of the network with values drawn from independent zero-mean Gaussian distributions with the variance decided by the standard fan-in and fan-out of the layer (He et al., 2015). $l = 1 \cdots L$ denotes the layer index. For each layer $l$, we pair $\boldsymbol{W}_l$ and $\boldsymbol{b}_l$ with learnable non-negative scalars $\alpha_i$ and $\beta_i$ that control the scaling of the layer at initialization, and we use $\boldsymbol{\beta}$ to denote the vector of scale factors $(\xi_1, \zeta_1, \cdots, \xi_L, \zeta_L)$, and let $\boldsymbol{\theta}^{\boldsymbol{\beta}}$ be the tuple $(\xi_1 \boldsymbol{W}_1, \zeta_1 \boldsymbol{b}_1, \cdots, \xi_L \boldsymbol{W}_L, \zeta_L \boldsymbol{b}_L)$ of rescaled parameters. Also, we extend the loss definition as

$$L(S; \boldsymbol{\theta}) = \frac{1}{|S|} \sum_{(\mathbf{x}, y) \in S} \ell(f(\boldsymbol{\theta}; \mathbf{x}), y), \tag{3}$$

which is the average loss of the model with $\boldsymbol{\theta}$ on a minibatch of samples $S$. Correspndingly, we denote one SFW optimization step obtained on minibatch $S$ as $\mathrm{SFW}(S, f, \mathcal{C}(\{K_l\}_{l=1}^L, \tau), \boldsymbol{\theta}_0, 1, \alpha_0, \rho)$.

---

[1]Here $K_l = 5\%$ refers to 5% of the weights in layer $l$. Also, we let $\tau$ be rescaled by the expected initialization norm. See Section 5 for detailed implementation setup.

[2]We refer to the nature sparsity of a neural network as the largest pruning ratio that keeps the test performance undamaged.

---

**Algorithm 3:** Stochastic Frank-Wolfe Initialization Scheme (SFW-Init)

---

1: **Input:** Dataset $\mathbb{D}$, SFW parameters (See Algorithm 1), $K$-sparse polytope constraint
$\quad\ \mathcal{C}(\{K_l\}_{l=1}^{L}, \tau)$, learning rate $\kappa$, total iterations $T$, lower bound of weight and bias scales $\epsilon, \varepsilon$.
2: **Output:** A new initialization $\boldsymbol{\theta}_0^{\boldsymbol{\beta}} = \texttt{SFWInit}(f(\boldsymbol{\theta}_0; \cdot), \boldsymbol{\theta}_0)$.
3: Set $\boldsymbol{\beta}_1 \leftarrow \mathbf{1}$
4: **for** $t = 1, \cdots, T$ **do**
5: $\quad$ Draw random samples $S_t$ from training set $\mathbb{D}$.
6: $\quad$ Draw $|S_t|/2$ samples to replace $|S_t|/2$ samples in $S_t$ and let this new minibatch be $\tilde{S}_t$.
7: $\quad$ Set $L_{t+1} \leftarrow \frac{1}{|\tilde{S}_t|} \sum_{(\mathbf{x},y) \in \tilde{S}_t} \ell(f(\texttt{SFW}(S_t, f, \mathcal{C}(\{K_l\}_{l=1}^{L}, \tau), \boldsymbol{\theta}_0^{\boldsymbol{\beta}_t}, 1, \alpha, \rho); \mathbf{x}), y)$.
8: $\quad$ Set $\tilde{\boldsymbol{\beta}}_{t+1} = \boldsymbol{\beta}_t - \tau \nabla_{\boldsymbol{\beta}} L_{t+1}(\boldsymbol{\theta}^{\boldsymbol{\beta}_t})$.
9: $\quad$ Clamp $\tilde{\boldsymbol{\beta}}_{t+1}$ using the lower bound $\epsilon$ and $\varepsilon$.
10: $\quad$ Project $\tilde{\boldsymbol{\beta}}_{t+1}$ back to $\mathcal{C}(\tau, K)$ as $\boldsymbol{\beta}_{t+1} = \Pi_{\mathcal{C}(\tau, K)}(\tilde{\boldsymbol{\beta}}_{t+1})$.
11: **end for**

---

Specifically, we optimize the scaling parameter $\boldsymbol{\beta}$ to obtain a good initialization $\boldsymbol{\theta}^{\boldsymbol{\beta}}$ tailored for a stochastic Frank–Wolfe algorithm, i.e., SFW. Consider the first update step, starting from the rescaled initialization, i.e., $\boldsymbol{\theta}_1 = \texttt{SFW}(S, f, \mathcal{C}(\{K_l\}_{l=1}^{L}, \tau), \boldsymbol{\theta}_0, 1, \alpha_0, \rho)$. We choose $\boldsymbol{\beta}$ so that the loss on $\boldsymbol{\theta}_1$ is as low as possible. Formally, we optimize the scaling $\boldsymbol{\beta}$ through the following objective:

$$\min_{\boldsymbol{\beta} \ s.t. \boldsymbol{\theta}_0^{\boldsymbol{\beta}} \in \mathcal{C}(K,\tau)} L\left(\tilde{S}; \texttt{SFW}(S, f, \mathcal{C}(\{K_l\}_{l=1}^{L}, \tau), \boldsymbol{\theta}_0, 1, \alpha_0, \rho)\right), \tag{4}$$

where $S$ and $\tilde{S}$ are two different random minibatches. For his paper, we only consider $K$-sparse polytope constraints, which can be easily extended to other constraints like $\ell^p$-norm balls. To make sure that the rescaled parameter $\boldsymbol{\theta}^{\boldsymbol{\beta}}$ is still a feasible initialization, we make the constraint that $\boldsymbol{\theta}^{\boldsymbol{\beta}} \in \mathcal{C}(K, \tau)$. We conclude the new initialization scheme for SFW in Algorithm 3. Our algorithm features the following points, which differ from the work of Zhu et al. (2021):

- We do not apply constraints on the norm of the gradient $\nabla_{\boldsymbol{\theta}} L_t(\boldsymbol{\theta}^{\boldsymbol{\beta}_t})$, which is the case in Zhu et al. (2021). This is because for $K$-sparse polytope constraints the linear minimization oracle $\texttt{LMO}_{\mathcal{C}(\{K_l\}_{l=1}^{L}, \tau)}(\nabla_{\boldsymbol{\theta}} L_t(\boldsymbol{\theta}^{\boldsymbol{\beta}_t}))$ only depends on the direction rather than norm.

- Since we need to ensure that the initialization is feasible, we project the obtained rescaled initialization $\boldsymbol{\theta}^{\boldsymbol{\beta}_t}$ back to the $K$-sparse polytope constraints at each training iteration. This is done via a simple scaling of the current parameter to the $K$-sparse polytope $\mathcal{C}(\tau, K)$.

Besides, there are two other key points in the SFW-Init algorithm, motivated by Zhu et al. (2021). The first point is that we use different but overlapping minibatches $S$ and $\tilde{S}$ to calculate the current loss and the first SFW step respectively. The difference between two mini-batches is to prevent overfitting, while the overlapping of the minibatches is for better train performance. See Zhu et al. (2021) for a study of $S$ and $\tilde{S}$. The second point is that we detach the SFW algorithm step from the gradient $\nabla_{\boldsymbol{\beta}} L_{t+1}$ (line 7) in order to facilitate the differentiation of the SFW algorithm (which involves sorting the coordinates of the gradient vector) and potential second-order derivatives. The corresponding SFW algorithm modification is marked in the grey color.

## 5 EXPERIMENTS

### 5.1 SETTINGS

We summarize the key experiment setups, with hyperparameters of the implementation presented in Appendix A in detail. We conduct experiments via two popular architectures, ResNet-18 (He et al., 2016) and VGG-16 (Simonyan & Zisserman, 2014), on two benchmark datasets, CIFAR-10 (Krizhevsky et al., 2009) and Tiny-ImageNet (Wu et al., 2017). Specifically, on all these setups, we demonstrate our SFW-pruning framework (Algorithm 2) by first solving the $K$-sparse polytope constrained optimization problem via SFW with momentum and then applying one-shot pruning to different sparsity ratios. We train each network for 180 epochs using a dynamic-changing learning

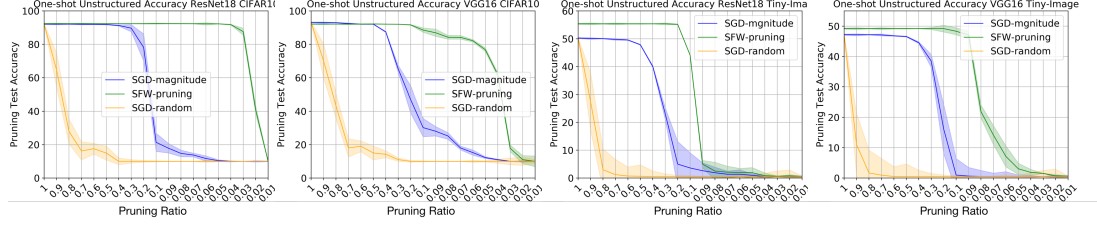

(a) CIFAR-10 ResNet-18    (b) CIFAR-10 VGG-16    (c) Tiny-Image ResNet-18    (d) Tiny-Image VGG-16

Figure 2: Test performance of unstructured pruning networks with different sparsity ratios *without retraining*. We compare three settings: SFW-pruning, SGD training with one-shot weight-magnitude pruning, SGD-training with one-shot random pruning. All initial learning rate is fixed to $\alpha_0 = 1.0$.

rate schedule used by Pokutta et al. (2020). We evaluate the neural networks by testing accuracy after pruning under different sparsity ratios and *without* retraining.

## 5.2 ONE-SHOT WEIGHT-MAGNITUDE UNSTRUCTURED PRUNING

We first evaluate our SFW-pruning framework in unstructured weight-magnitude pruning paradigm (Han et al., 2015; Frankle & Carbin, 2018) across a total of four combinations of datasets and architectures, i.e., CIFAR-10 with {ResNet-18, VGG-16} and Tiny-ImageNet with {ResNet-18, VGG-16}. Note that for the fairness of comparison, we **do not** use our learned initialization (SFW-Init) here.

Specifically, after training the DNN with $K$-sparse polytope constraints via SFW, we prune it to various sparsity ratios by leaving the largest weights unchanged and the smaller ones pruned. We consider neural network training without constraints via SGD and random one-shot pruning for comparison. Figure 2 shows the achieved test performance of the pruned neural networks with different sparsities. Several observations can be drawn from Figure

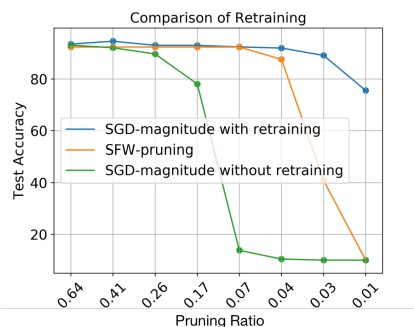

Figure 3: Our one-shot SFW-pruning (no retraining) can even achieve competitive performance against "SGD-pruning with extra retraining costs" (ResNet-18, CIFAR-10).

2: (i) without retraining, SFW-pruning significantly outperforms magnitude-based and random pruning by SGD, across all architectures and datasets; (ii) over a wide range of sparsity ratios, SFW can keep pruning while maintaining a highly competitive performance without any retraining cost.

**SFW-pruning is Competitive Over "SGD + Retraining".** We also compare our SFW-pruning to SGD-pruning *with* retraining, which is to our disadvantage. In Figure 3, SFW-pruning can achieve comparable accuracies against SGD-pruning with extra retraining cost, across most pruning ratios. In contrast, without retraining, SGD-pruning can only tolerate pruning ratios above 30%. This further validates that SFW-pruning is sparsity-aware. Network weights of less importance have already converged to small magnitudes by SFW. Unlike the performance gap induced by hard-thresholding by SGD, important weights found and trained by SFW-pruning are ready for pruning and inference.

## 5.3 ADDING THE NEW SFW INITIALIZATION

We then evaluate the effectiveness of the initialization scheme (SFW-Init) designed specifically for SFW algorithm in Section 4. We repeat the previous experiments for evaluating SFW-pruning framework in an unstructured paradigm, comparing the test performance with and without the new initialization scheme. The corresponding results are shown in Figure 4. We can see that the performance after adopting our customized SFW initialization is consistently better than Kaiming initialization.

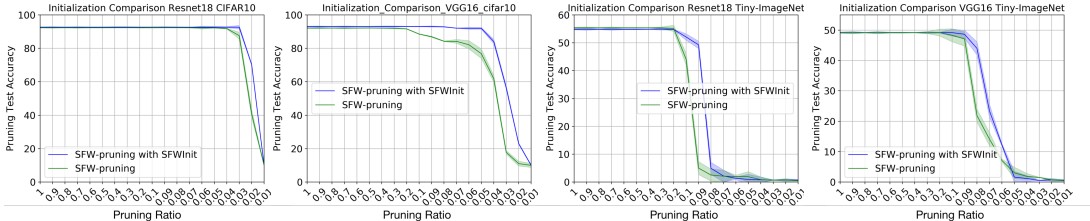

| (a) CIFAR-10 ResNet-18 | (b) CIFAR-10 VGG-16 | (c) Tiny-Image ResNet-18 | (d) Tiny-Image VGG-16 |

Figure 4: We study SFW-pruning with and without `SFWInit`. Test accuracies of the pruned DNNs with different sparsity ratios are obtained *without retraining*. All have initial learning rate $\alpha_0 = 1.0$.

## 5.4 COMPARISON TO STATE-OF-THE-ART METHODS

We now compare our holistic solution: "`SFW-pruning + SFW-Init`", with other state-of-the-art (SOTA) unstructured pruning methods (Lee et al., 2019; Blalock et al., 2020; Tanaka et al., 2020). Note that most previous works require either iterative pruning or retraining for each sparsity ratio, which costs extremely heavy overhead and is to our disadvantage. The full comparison of VGG-16 on CIFAR-10 is demonstrated in Figure 5. Compared with both iterative pruning and pruning at initialization approaches, our method generates sparse networks at a range of pruning ratios in one-shot fashion that all maintain highly competitive performance.

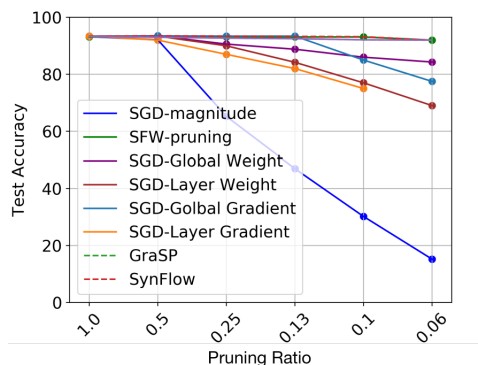

Figure 5: Comparison with SOTA unstructured pruning methods on VGG-16 and CIFAR-10. Solid lines are reported by Blalock et al. (2020).

We further compare with the more recent "one-shot no-retrain" method (Chen et al., 2021), pruning-during-training methods (You et al., 2020; Lin et al., 2020; Hubens et al., 2021), and group-sparsity inducing method (Deleu & Bengio, 2021). As demonstrated in Table 1, our method outperforms all others with the only exception of DPF (Lin et al., 2020). However, DPF requires full re-training for each pruning ratio, whereas ours only trains once for all possible ratios.

| **Pruning Ratios** | **50%** | **70%** | **80%** | **90%** | **95%** |
|---|---|---|---|---|---|
| SFW-Pruning + SFW-Init (ours) | 93.10 | 93.10 | 93.10 | 93.10 | 92.00 |
| One-Cycle Pruning (Hubens et al., 2021) | - | - | 90.87 | 90.72 | 90.67 |
| Early Bird (You et al., 2020) | 93.21 | 92.80 | - | - | - |
| OTO (Chen et al., 2021) | 90.35 | 90.35 | 90.35 | 90.35 | 90.35 |
| DPF (Lin et al., 2020) | - | - | - | - | 93.87 |
| Group MDP (Deleu & Bengio, 2021) | - | - | - | 89.38 | - |

Table 1: Comparisons to more state-of-the-art methods on VGG-16 and CIFAR-10.

## 6 CONCLUSION

We proposed to consider Stochastic Frank-Wolfe (SFW) for train pruning-friendly networks. Unlike previous pruning methods based on SGD that often require (iterative) pruning followed by retraining, our method can prune a network to any sparsity by training only once, which largely eliminates the pruning overhead. Our core contribution is to train the network weights within in a convex hull spanned by $K$-sparse vectors, yielding much more smaller weights yet meanwhile "smoother" magnitude distribution, that is more amendable to gradual weight removal towards any sparsity ratio. Our work is a pilot study to demonstrate the potential of non-gradient optimization methods in training deep networks, especially when additional weight structures like sparsity are desired.

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

# A    IMPLEMENTATION DETAILS

Our experiments are conducted with ResNet-18 (He et al., 2016) and VGG-16 (Simonyan & Zisserman, 2014) on two benckmark datasets, CIFAR-10 (Krizhevsky et al., 2009) and Tiny-ImageNet (Wu et al., 2017). We first conclude all hyperparameters of SFW-pruning in the following table.

| Hyperparameters | Value | Hyperparameters | Value | Hyperparameters | Value |
|---|---|---|---|---|---|
| Initial learning rate $\alpha_0$ | 1.0 | Training batchsize | 128 | Test batchsize | 100 |
| Radius $\tau$ | 15 | $K$-frac $\{K_l\}_{l=1}^L$ | 5% | Training epoch $T$ | 180 |
| Momentum $\rho$ | 0.9 | | | | |

Table 2: Values of hyperparameters in SFW-pruning (Algorithm 2).

Moreover, there are two other important technical points in choosing the learning rate $\alpha_t$ during SFW training (Pokutta et al., 2020) that we adopt in our experiments. We clarify them in the following.

- The first point is the learning rate changing scheme. We decrease the learning rate by 10 at epoch 61 and 121. Also, we *dynamically* change the learning rate (Pokutta et al., 2020): the learning rate is multiplied by 0.7 if the 5-epoch average loss is greater than the 10-epoch average loss, and is increased by a factor 1.06 if the opposite holds.

- The second point is to rescale the effective learning rate. To be specific, in order to make tuning of the learning rate easier (Pokutta et al., 2020), one can decouple the learning rate from the size of the feasible region. Here we adopt the following modified update rule,

$$\theta_{t+1} = \theta_t + \min\{\alpha_t \|\widehat{\nabla} L(\theta_t)\|_2 / \|v_t - \theta_t\|_2, 1\} (v_t - \theta_t). \tag{5}$$

Finally, we conclude the hyperparameters of SFWInit in the following table.

| Hyperparameters | Value | Hyperparameters | Value | Hyperparameters | Value |
|---|---|---|---|---|---|
| Learning rate $\kappa$ | 0.001 | Training iterations $T$ | 390 | Minimal scaling $\epsilon, \varepsilon$ | 0.01 |

Table 3: Values of hyperparameters in SFWIint (Algorithm 3).

# B    $K$-SPARSE POLYTOPE CONSTRAINTS AND DNN WEIGHT DISTIRBUTIONS

In this section, we give a detailed description of the $K$-sparse polytope constraint that we use in our algorithm. In Section B.1, we first give two equivalent strict definitions of a $K$-sparse polytope and we further explain the motivation why it benefits to equip such a constraint in our training objective. Then in Section B.2, we analysis the choice of the parameters in the $K$-sparse polytope constraint via ablation studies on how they influence the pruning performance and the weight distributions. The corresponding results further demonstrate our motivation of using such a constraint. Finally, in Section B.3, we show the weight distributions induced by SFW-training and SGD-training across different architectures and datasets which demonstrate the universal capability of encouraging sparse weights of SFW-training.

## B.1    MORE ABOUT $K$-SPARSE POLYTOPE CONSTRAINTS

A $K$-sparse polytope in $\mathbb{R}^p$ of radius $\tau > 0$, $p \in \mathbb{N}$, is a bounded subset of $\mathbb{R}^p$, which is denoted by $\mathcal{C}(K, \tau)$. There are two equivalent ways to define a $K$-sparse polytope $\mathcal{C}(K, \tau)$.

1. A $K$-sparse polytope $\mathcal{C}(K, \tau)$ is the intersection of the $L^1$-ball $\mathcal{B}_1(\tau K)$ and the $L^\infty$-ball $\mathcal{B}_\infty(\tau)$ of $\mathbb{R}^p$, i.e., $\mathcal{C}(K, \tau) = \{\boldsymbol{x} \in \mathbb{R}^p : \|\boldsymbol{x}\|_1 \leq \tau K, \ \|\boldsymbol{x}\|_\infty \leq \tau\}$.

2. A $K$-sparse polytope $\mathcal{C}(K, \tau)$ can also be obtained by spanning all the vectors in $\mathbb{R}^p$ which have exactly $K$ non-zero coordinates and the absolute value of the non-zero entries are $\tau$, i.e., $\mathcal{C}(K, \tau) = \text{Span}_{[0,1]}(\{\boldsymbol{v} \in \mathbb{R}^p : \|\boldsymbol{v}\|_0 = K, \ (\boldsymbol{v})_i \in \{0, \tau\}\})$.

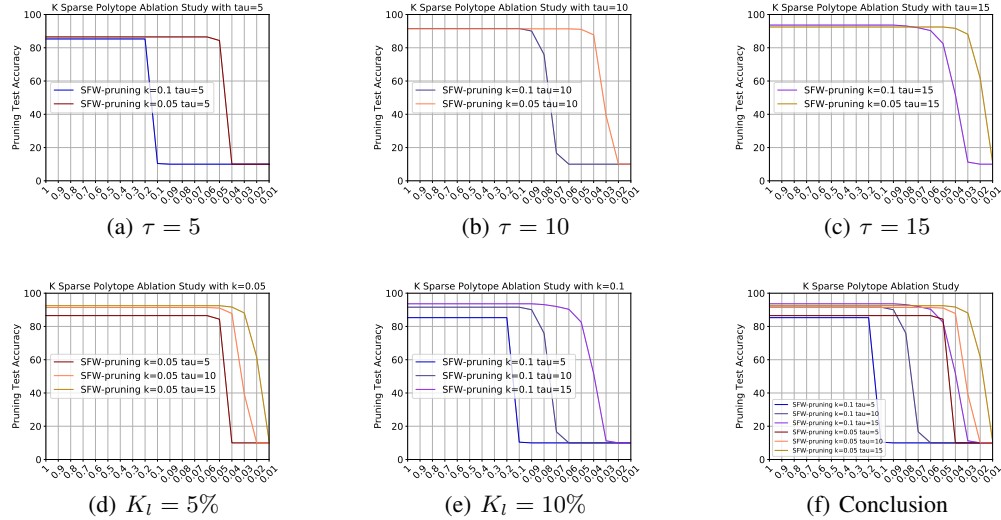

Figure 6: Ablation study of the influence of $\tau$ and $K$ in $K$-sparse polytope constraint on the pruning performance (ResNet18 on CIFAR-10). We consider $\tau \in \{5, 10, 15\}$ and $K_l \in \{5\%, 10\%\}$.

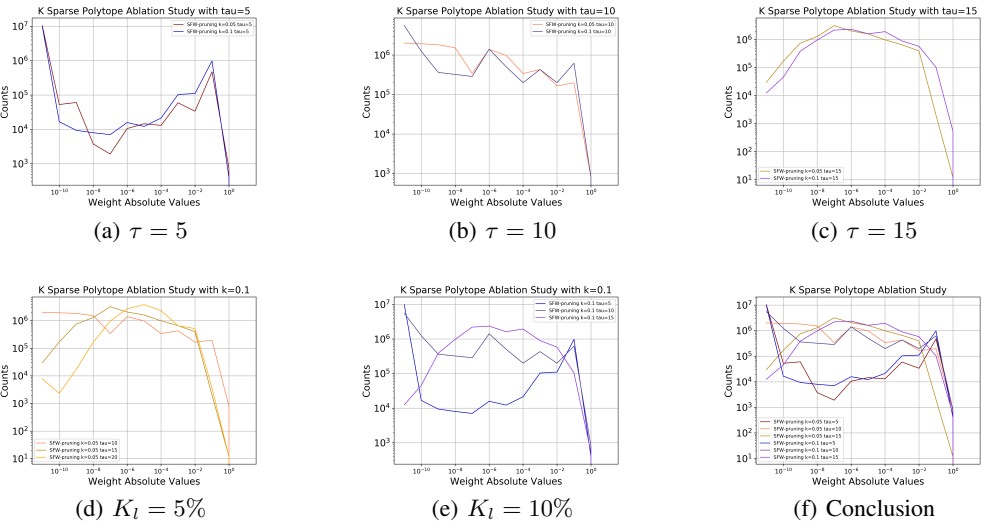

Figure 7: Ablation study of the influence of $\tau$ and $K$ in $K$-sparse polytope constraint on the weight distributions (ResNet18 on CIFAR-10). We consider $\tau \in \{5, 10, 15\}$ and $K_l \in \{5\%, 10\%\}$.

It holds that (Pokutta et al., 2020) for any $\boldsymbol{m} \in \mathbb{R}^p$ the oracle $\boldsymbol{v} = \text{LMO}_{\mathcal{C}(K, \tau)}(\boldsymbol{m})$ is given by

$$(\boldsymbol{v})_i = \begin{cases} -\tau \cdot \text{sign}((\boldsymbol{m})_i) & \text{if } (\boldsymbol{m})_i \text{ is in the largest } K \text{ coordinates of } \boldsymbol{m}, \\ 0 & \text{otherwise,} \end{cases} \quad \forall 1 \leq i \leq p, \quad (6)$$

which is a vector with exactly $K$ non-zero entries. According to the SFW update formula $\boldsymbol{\theta}_{t+1} = \boldsymbol{\theta}_t + \alpha_t(\boldsymbol{v}_t - \boldsymbol{\theta}_t) = \alpha_t \boldsymbol{v}_t + (1 - \alpha_t)\boldsymbol{\theta}_t$, the DNN weight $\boldsymbol{\theta}_t$ is averaged with the $K$-sparse vector $\boldsymbol{v}_t$. Thus each SFW step pushes those less important weights smaller since they are averaged with zero, and those more important weights are enhanced. The motivation is that this can induce a weight distribution of DNN that has large amounts of small weights. With such a weight distribution, one-shot pruning can more smoothly remove small values when the pruning ratio increases, yielding competitive test accuracy across the spectrum of pruning ratios, even when retraining is prohibited.

## B.2 ABLATION STUDY OF $K$-SPARSE POLYTOPE CONSTRAINTS PARAMETERS

**Ablation Study of $K$-sparse Polytope Constraint Hyperparameters.** We now conduct ablation studies on how the choices of $\tau$ and $K$ in the $K$-sparse polytope constraints can influence the test performance of SFW-pruning algorithm and the weight distributions of the trained DNNs. We run the SFW-pruning algorithm (Algorithm 2) using ResNet18 on CIFAR-10 with a range of different hyperparameters. Specifically, we consider $\tau \in \{5, 10, 15\}$ and $K_l \in \{5\%, 10\%\}$. Figure 6 shows the ablation study of $\tau$, $K$ choices on the pruning test performance. Figure 7 shows the ablation study of $\tau$, $K$ choices on the weight distributions of the trained DNNs.

From (a) to (d) in Figure 6 and 7 we can find that with *smaller $K$*, SFW-training can induce weight distributions with more smaller weights, which at the same time gives better pruning performance across different pruning ratios. But this is at the cost of *lower test performance of the full model*. This observation matches our motivation for using $K$-sparse polytope constraint. When $K$ is smaller, the update becomes sparser (recall the LMO solution (6)), which means that more weights are averaged with zero. Consequently, the final weight distributions will gain more smaller weights, which further benefits the pruning procedure since we can more smoothly remove small values when the pruning ratio increases without harming the test performance.

From (e) to (g) in Figure 6 and 7, we can see that with smaller $\tau$, SFW-training can induce weight distributions with more smaller weights. But these weight distributions with too many smaller weights do not always generate better pruning performance consistently, e.g., see (f) in Figure 6 and 7.

Given the above ablation study we choose $\tau = 15$ and $K_l = 5\%$ as the hyperparameters in our experiments. This can give both satisfactory full model performance and pruning performance across different pruning ratios. We note that by choosing smaller $K_l$, e.g., $K_l = 1\%$ one can get even better pruning performance, but the full model performance also suffers.

**Ablation Study of Initial Learning Rate.** Moreover, we study the influence of initial learning rate $\alpha_0$ on the test performance of SFW-pruning algorithm and the weight distributions of the trained DNNs. Specifically we run the SFW-pruning algorithm (Algorithm 2) using VGG16 on CIFAR-10 with a range of initial learning rate $\alpha_0 \in \{0.1, 0.2, 0.4, 0.6, 0.8, 1.0\}$. Figure 8 shows the corresponding pruning performance and weight distributions. One can see that larger learning rates encourage smaller weights and better performance (on both small and large pruning ratios).

Our conclusion still holds on more complex datasets like TinyImageNet. As is shown in Figure 9, a weight distribution with more smaller values is still more friendly to pruning.

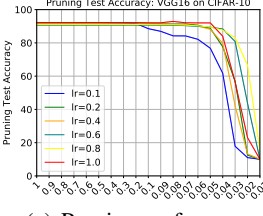
(a) Pruning performance

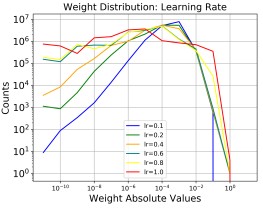
(b) Weight Distributions

Figure 8: Comparison of pruning performance and weight distributions induced by SFW-pruning for VGG16 on CIFAR-10 with different initial learning rates $\alpha_0 \in \{0.1, 0.2, 0.4, 0.6, 0.8, 1.0\}$

## B.3 PERFORMANCE ON MORE ARCHITECTURES AND DATASETS

Finally, we show the weight distributions induced by SFW-pruning across four different architecture and dataset combinations, and we compare them to the weight distributions induced by SGD training as well. Specifically, we consider {ResNet18, VGG16} and {CIFAR-10, CIFAR-100} respectively. The results are summarized in Figure 10.

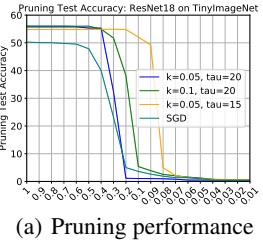 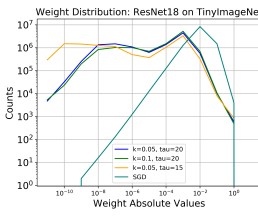

(a) Pruning performance         (b) Weight distributions

Figure 9: Comparison of pruning performance and weight distributions induced by SFW-pruning for ResNet18 on TinyImageNet. The results show that even on complex dataset like TinyImageNet, appropriately more smaller weights can still benefit the pruning procedure.

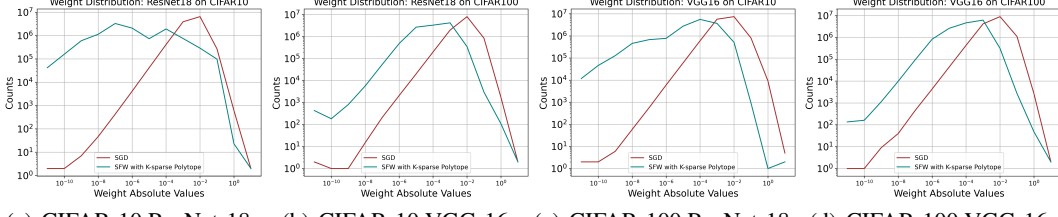

(a) CIFAR-10 ResNet-18  (b) CIFAR-10 VGG-16  (c) CIFAR-100 ResNet-18  (d) CIFAR-100 VGG-16

Figure 10: Comparison of weight distributions induced by SFW and SGD over different architectures and datasets. For SFW-pruning, we choose $\tau = 15$ and $K_l = 5\%$, with $\alpha_0 = 1.0$.

