# OpenReview forum: "Learning Pruning-Friendly Networks via Frank-Wolfe: One-Shot, Any-Sparsity, And No Retraining"
_ICLR.cc/2022/Conference — ICLR 2022 Spotlight_

### Official Review · Reviewer_urBu · 2021-10-31

**Correctness:** 3
**Technical Novelty And Significance:** 3
**Empirical Novelty And Significance:** 3
**Recommendation:** 6
**Confidence:** 4

**Main Review:**

The core strength of this paper is to train a deep neural network so that it is more resilient to pruning. The author achieves this by using the K-sparse polytope constraint to enforce the model has more small weights. And they empirically show that such weight distribution can help weight pruning. They also use stochastic Frank-Wolfe for optimizing the problem. A gradient-based initialization scheme is also used to improve the trade-off between sparsity and performance.

The weakness of this paper can be summarized into the following points:
1. The core benefit of the proposed method is not clear.  In 'Why K-Sparse Polytope Constraints?', authors argue that 'each SFW step pushes those less important weights smaller' and such weight distribution 'can more smoothly remove small values' and 'yielding competitive test accuracies'. On the other hand, in 'Learning rate $\alpha$', authors argue that a larger learning rate can 'result in sparser parameter $\theta_{t+1}$' and higher natural sparsity. The latter argument suggests that the benefit of K-sparse constraints is due to they can learn sparser solutions for the deep neural network. The former argument suggests that the benefit of K-sparse constraints is a weight distribution that can make pruning easier. The latter argument is a quite trivial result. The former argument is interesting but requires more analysis. In addition, it seems that the K-sparse constraint is not designed to obtain a sparse solution. The property of the Frank-Wolf algorithm ensures $\theta$ always attains the K-sparse constraint. If the K-sparse constraint is really useful, then it is not clear why $\alpha$ is important to the performance.

2. The authors should provide a more detailed analysis of K and $\tau$ for K-sparse polytope constraints, and how they affect the weight distributions. I also suggest the authors include a more detailed definition of K-sparse polytope constraints and their properties, to make the paper self-contain.

3. Authors use the Frank-Wolfe framework, which is drastically different from regular techniques for training deep neural networks. As a result, ImageNet experiments with standard baseline models, like ResNet-50, are required, and numerical results should also be reported. The reason is that scalability is crucial to weight pruning algorithms. Moreover, on ImageNet, whether Frank-Wolfe algorithms can obtain similar performance as popular stochastic optimizers is not clear.

4. In Figure.2, the advantage of the proposed method becomes much smaller when the dataset is more complex. When applying large models on small datasets, it is plausible to say small weights are not important. However, when the dataset becomes complex, small weights may also become important as suggested by previous works [1]. From this perspective, the effects of pushing more weights smaller may vanish when the dataset is large enough.

[1] Ye, Jianbo, et al. "Rethinking the smaller-norm-less-informative assumption in channel pruning of convolution layers." ICLR, 2018.

----------------------------------------------------

After the authors' response, most of my questions were addressed, and I increased my score to 6.


**Summary Of The Paper:**

In this paper, the authors proposed to use stochastic Frank-Wolfe with K-sparse polytope constraints to train deep neural networks and make them pruning friendly.

**Summary Of The Review:**

In summary, I think this paper is below the acceptance threshold since the main claim lacks enough support. And current results may be due to sparsity incurred by the large learning rate $\alpha$. In addition, experimental results are not enough.

---

> ### Author Response · Authors · 2021-11-19
> **Response to Questions from Reviewer urBu**
>
> We appreciate reviewer urBu's important questions and address them below.
>
> **1. More analysis on $\tau$, K, and learning rate (Q1 and Q2)**
>
> We address your concerns in two aspects:
>   - (1) On the empirical level, we further conduct a more detailed ablation study on different $\tau$, K, and learning rate in Appendix B.3. The conclusions are:
>
> (i) for pruning performance, a larger $\tau$, a smaller K, and a large learning rate ($\alpha$) are better (Fig 6);
>
> (ii) for weight distribution, a larger $\tau$ encourage a more continuous change on the number of weights from small values to large values, whereas a small $\tau$ incurs two peaks (Fig. 7a), leaving a gap between the number of large weights and small weights and being harmful to network training. Meanwhile, a smaller K and a larger learning rate $\alpha$ will shift the weight distribution to the left, being more friendly to pruning.
>
> These conclusions lead us to choose K = 5% and $\tau$ = 15, which performs the best. Further, a large learning rate as 1 performs the best under both small and large pruning ratios.
>
>   - (2) On the methodology level, SFW could be considered as: keep the k-largest elements in gradient, only the sign but not magnitude. Hence our understanding is: this is becoming like a "voting" process, where each minibatch has K votes to boost K out of all weights, and all "votes" now are equal (not by magnitude). In this way, the total weight update is becoming a "histogram" of how many mini-batches vote on what weight. This process "erases out" the noisy smaller magnitudes in each gradient and can also be more "fair" to avoid some larger updates dominating by having all non-zero votes equal like gradient clipping.
>
> Furthermore, we want to point out that "larger alpha more sparse" is correct because weight is in the span of the K-sparse vector, but not itself to be K-sparse. The K-Sparse polytope is the convex hull spanned by all vectors in $R^n$ with exactly K non-zero entries. And if smaller alpha, then the weight is closer to a smooth moving average of many K-sparse vectors of different supports (hence denser). If larger alpha, then this moving average is decaying very fast, and the current K-sparse update takes a more significant influence in shaping the next weight. Hence we think "The latter argument suggests that the benefit of K-sparse constraints is due to that hey can learn sparser solutions for the deep neural network" is not "quite trivial." Because only the gradient is K-sparse, weight is a stack of K-sparse but itself not.
>
> In summary,  a larger $\alpha$ tends to speed up the k sparse moving average, resulting in more small weights. And the core benefit of using the K-Sparse polytope constraint is that it can naturally generate a smooth weight distribution that can be easily pruned.
>
> **3. About “more detailed definition”**
>
> Thank you for the suggestions! We have revised our draft with more detailed definitions and descriptions of K-sparse polytope and its properties in Appendix B.2.
>
> **4. ImageNet experiments.**
>
> Thanks for this suggestion! We are currently running, and due to the limited rebuttal time window, we promise to include it in the final version.
>
> **5. Small weights still matter on complex datasets**
>
> We address your concern in two aspects:
>   - (1) On the empirical level, we further conduct an ablation study on TinyImageNet, in Figure in Appendix B.2. Again, after comparing with SGD and different K and \tau, we can see that a weight distribution shift to left (small values) will lead to better pruning performance. This validates that small weights are still important for pruning on a more complex dataset.
>   - (2) On the methodology level, it is true that small weights also matter, and meanwhile, more recent findings demonstrate magnitude-based are surprisingly strong and outperform all gradient/other score-based methods even on sufficiently complex models and datasets, when tuned right, see: https://arxiv.org/pdf/2003.03033.pdf. Hence, we think grounding SFW-pruning on weight magnitude is solid, and we will also add your suggestions in the revision. It might be true there might be better than the magnitude pruning method, but they are out-of-scope.

---

### Official Review · Reviewer_22uh · 2021-11-02

**Correctness:** 3
**Technical Novelty And Significance:** 3
**Empirical Novelty And Significance:** 3
**Recommendation:** 6
**Confidence:** 3

**Main Review:**

The topic studied by this paper is of general interest. The authors manage to organize the paper in a good way.

Strength:

(1) Formulate the pruning strategy with an optimization problem and solve it through Frank-Wolfe

(2) The learning-based initialization helps to achieve better performance.

Questions:

(1) Which step in the proposed algorithm other than backpropagation has the largest cost per iteration?

(2) Does the algorithm model-agnostic? How is the performance in various vision models

(3) Does the proposed algorithms scales to larger dataset such as ImageNet?

**Summary Of The Paper:**

This paper proposes a Frank-Wolfe based approach for efficient pruning.
The highlights of this paper are:

(1)  Propose a learning-based initialization scheme for SFW-based DNN training

(2)  Achieves promising performance on DNN benchmark when we set different pruning ratios

**Summary Of The Review:**

The paper proposes an interesting algorithm for training large DNNs. Particularly, the proposed method could handle different pruning ratios.
There still exist concerns regarding the scalability of the algorithm. The authors are encouraged to address it with more experiments.

---

### Official Review · Reviewer_BXpo · 2021-11-03

**Correctness:** 3
**Technical Novelty And Significance:** 4
**Empirical Novelty And Significance:** 3
**Recommendation:** 8
**Confidence:** 4

**Main Review:**

# Strengths

1. The problem of pruning-aware training is interesting and has a lot of potential.
2. The formulation and approach to optimize it using SFW is novel.
3. In addition an initialization scheme that is suited for SFW is also presented.
4. The results demonstrate the value of pruning-aware training using the proposed approach.

# Weaknesses

1. The motivation for K-sparse constraint set is not clear. More clarity is required on how the authors chose to use such a constraint set.

2. If I understand correctly, the \tau is a constant for all the weights in the network and therefore, the SFW update at any iteration is basically based on the sign of the gradient. Is this understanding correct? If yes, it further raises questions about the choice of the K-sparse constraint set. What is the need to use such a fixed \tau? How to decide the value of \tau? It is intriguing that even with such a restrictive update the final weights are competitive to SGD based training.

3. The performance is still worse than SGD with retraining at extreme sparsities (fig. 5) but considering the cost of retraining the proposed method is useful.

**Summary Of The Paper:**

The paper presents an algorithm to train a DNN in a pruning-aware manner such that the trained network can be pruned at various sparisities without any fine-tuning. The approach is based on stochastic frank-wolfe and the results are impressive for a wide-range of sparsities.

**Summary Of The Review:**

The idea is interesting and the method is technically sound. The choice of the constraint set should be motivated and clarified to improve the quality of the paper.

---

### Official Review · Reviewer_9FPN · 2021-11-03

**Correctness:** 3
**Technical Novelty And Significance:** 2
**Empirical Novelty And Significance:** 2
**Recommendation:** 5
**Confidence:** 4

**Main Review:**

I think the proposed method here is actually quite interesting but this work fails to discuss a whole body of relevant literature that also aims to train sparse neural networks. As a result the chosen baselines for the experiments section fail to show how effective this method actually is.

Here are few examples I was expecting to see discussed and/or evaluated:

- Directly related to the "one-shot no retrain" claim of this paper:
    - [Only Train Once: A One-Shot Neural Network Training And Pruning Framework by Chen et al](https://arxiv.org/abs/2107.07467)
- Pruning during training:
    - [Sparse Networks from Scratch: Faster Training without Losing Performance by Dettmers et al.](https://arxiv.org/abs/1907.04840)
    - [The State of Sparsity in Deep Neural Networks by Gale et al.](https://arxiv.org/abs/1902.09574)
    - [Dynamic Model Pruning with Feedback by Lin et al.](https://arxiv.org/abs/2006.07253)
- there are also lots of works on inducing group sparsity on neural networks that's relevant to this paper. Some examples are:
    - [Towards Compact ConvNets via Structure-Sparsity Regularized Filter Pruning by Lin et al.](https://arxiv.org/abs/1901.07827)
    - [Learning Structured Sparsity in Deep Neural Networks by Wen et al.](https://arxiv.org/abs/1608.03665)
    - [Structured Sparsity Inducing Adaptive Optimizers for Deep Learning by Deleu et al.](https://arxiv.org/abs/2102.03869)
    - [Neuron-level Structured Pruning using Polarization
Regularizer by Zhuang et al.](https://papers.nips.cc/paper/2020/file/703957b6dd9e3a7980e040bee50ded65-Paper.pdf)
- and of course numerous papers that traine sparse neural networks with Lasso (or directly L0) regularization

Results in Figure 2:
- while it's true that SFW maintains performance in low pruning ratios but it seems to degrade a lot above 0.9 pruning ratio which is what typical pruning methods usually aim for. So doesn't it make more sense to compare SFW against other pruning methods at _their_ "nature sparsity"? I guess my point here is: why should I choose SFW over another classic pruning method that can achieve better performance at higher pruning ratio?
- In Fig2. the performance of SGD on TinyImageNet is lower than what it should be (and of course lower than Franke-Wolfe training). I was wondering if authors have an explanation for this?

Results in Figure 5:
- Section 5.4 is supposed to compare against SOTA pruning methods but chooses pruning-at-initialisation baselines such as GraSP and SynFlow. This makes very little sense to me. Why not compare against pruning-during-training methods or even pruning-after-training methods?

Random:
- Fig1a averages over multiple architectures and multiple datasets. Why does it make sense to do so? Is the shaded area the standard deviation over these differences or over multiple runs with different seed?
- Isn't Fig1.b exactly the same as Fig2.a?



**Summary Of The Paper:**

This paper swaps gradient-based optimisation for training neural networks with Frank-Wolfe algorithm with a constrain that pushes less important weights towards smaller values. The resulting trained models can be pruned to different sparsity targets without needing to be retrained.

**Summary Of The Review:**

This paper proposes an interesting alternative to gradient descent that can be used to train models with a weight distribution suitable for one-shot pruning. I find the method section well-written and convincing however I was expecting to see many other relevant work here and the experiments section fails to compare to these relevant works, so it's difficult for me to evaluate the effectiveness of the method compared to simpler/other alternatives.

---

> ### Author Response · Authors · 2021-11-27
> **Revisions and more comparisons are available. Look forward to more discussions.**
>
> Dear reviewer 9FPN:
>
> We appreciate all your suggestions and questions, and we responded in detail.
>
> Specifically, we added more comparisons with other pruning methods in the Table in our response below.
>
> A revised submission (PDF) is also available. In the appendix, We added more ablation study results and analysis on hyperparameters and weight distributions, and also more clear discussions on K-sparse polytope.
>
> We would appreciate it if you could please take a look and finalize your review on our work, hopefully more positively. We appreciate again for your valued efforts!
>
> Sincerely,
>
> Paper1627 Authors

---

### Author Response · Authors · 2021-11-23
**Revisions and responses are available. Look forward to more discussions.**

Dear reviewers and AC:

* A revised submission (PDF) is now available. In the appendix, We added more ablation study results and analysis on hyperparameters and weight distributions, and also more clear discussions on K-sparse polytope.
* We thank all suggestions and questions from reviewers, and we responded in detail. Specifically, we added more comparisons with other pruning methods, and included more discussions on the relationship between pruning performance and weight distributions.

We would appreciate it if reviewers could please take a look and finalize their reviews on our work, hopefully more positively. We thank everyone again for the valued efforts!

Sincerely,
Paper1627 Authors

---

### Author Response · Authors · 2021-11-29
**Summary of updates from Authors**

Dear reviewer and AC panel:

Thank you very much again for all your detailed reviews that have helped us revise our manuscript. We are glad to see that the novelty and empirical strength of our proposed pruning framework have been recognized by most reviewers. We also gently remind that the final decision time is only one day away and we are yet to hear from a few reviewers, particularly reviewer 9FPN, as we have tried to address their concerns about the value and effectiveness of our work given existing relevant literature. Now we summarize the major points of the paper (post-rebuttal) for a quick understanding of all:

**1. Reasons for choosing SFW-pruning**

We have highlighted that our SFW method should be chosen under a limited training cost and time budget since the core advantage of our method is that we only need one-time training to deliver N networks at N pruning ratios ready for inference. Beyond this saving of training cost, even without retraining, our methods can still achieve competitive performance at high pruning ratios.

**2. Concerns of effectiveness compared with existing works**

We added more comparisons with other baselines ([1]-[5]) of pruning with training in order to show the effectiveness of our method. Our SFW method can outperform most other works, and at the same time we only train once for all ratios.

**3. Clarification and motivation for K-sparse polytope constraint**

We have further elaborated in detail why we choose $K$​-sparse polytope constraint in our objective. Optimized by SFW, we achieve sparse gradient updates via using such constraints. By pushing those less important weights smaller and enhancing those more important weights, one-shot pruning by SFW can more smoothly remove small values as the pruning ratio increases without introducing performance gaps. Besides, we have added detailed and strict description and analysis of such constraints to make the paper self-contained.

**4. Concerns of hyperparameter choice**

We have added detailed ablation study on how the hyperparameters $\tau$, $K$ and $\alpha_0$ can influence the learned weight distributions and the corresponding pruning performance, which we also made more explanations on. Empirical findings in the ablation study demonstrate our motivation for using $K$-sparse polytope constraints (which enforce sparse gradient updates) in the objective.

**5. Concerns of scalability and the importance of small weights**

We have added more detailed study of SFW-pruning method in the Tiny-ImageNet dataset and have shown the importance of smaller weights. We are currently conducting experiments on the ImageNet dataset, and due to the limited rebuttal time window, we promise to include it in the final version.

**6. Clarification in understanding plots and a few writing concerns**

We have clarified the plot where we recognized a potential misunderstanding (Fig. 1.a in particular) might have happened. We make changes and decouple them into four individual figures in Appendix B.4. At the same time, several concerns about results in Figure 2 and Figure 5 have been discussed in the previous reply and our revision. The figures of ablation study results and corresponding analysis are also in Appendix B.



Finally, given the novelty and practical effectiveness of the work as already recognized by most reviewers, we truly look forward to possible discussions for more clear and fruitful resolution of the concerns highlighted above. Please do not hesitate to contact us if there are other clarifications or experiments we can offer. Thanks!

Best wishes,

Authors

Reference:

[1] Hubens, Nathan, Matei Mancas, Bernard Gosselin, Marius Preda, and Titus Zaharia. "One-cycle pruning: Pruning convnets under a tight training budget." arXiv preprint arXiv:2107.02086 (2021).

[2] You, Haoran, Chaojian Li, Pengfei Xu, Yonggan Fu, Yue Wang, Xiaohan Chen, Richard G. Baraniuk, Zhangyang Wang, and Yingyan Lin. "Drawing early-bird tickets: Towards more efficient training of deep networks." arXiv preprint arXiv:1909.11957 (2019).

[3] Chen, Tianyi, Bo Ji, Tianyu Ding, Biyi Fang, Guanyi Wang, Zhihui Zhu, Luming Liang, Yixin Shi, Sheng Yi, and Xiao Tu. "Only Train Once: A One-Shot Neural Network Training And Pruning Framework." arXiv preprint arXiv:2107.07467 (2021).

[4] Lin, Tao, Sebastian U. Stich, Luis Barba, Daniil Dmitriev, and Martin Jaggi. "Dynamic model pruning with feedback." arXiv preprint arXiv:2006.07253 (2020).

[5] Deleu, Tristan, and Yoshua Bengio. "Structured Sparsity Inducing Adaptive Optimizers for Deep Learning." arXiv preprint arXiv:2102.03869 (2021).

---

### Decision · Program_Chairs · 2022-01-20

**Decision:**

Accept (Spotlight)

**Comment:**

This work considers one-shot pruning in deep neural networks. The main departure from previous work is to consider stochastic Frank-Wolfe. The reported results are convincing although a number of baselines were missing from the initial submission. The authors provide a balanced account of the strengths and weaknesses of the proposed approach.

The authors adequately addressed the concerns of the reviewers. For instance they ran additional experiments to compare to missing pruning baselines. I would encourage the authors to revise the manuscript by including the missing related work, the additional clarification discussions (e.g., motivation for K-sparse constraints, follow-up analysis, and cost per iteration) and to include the additional experiments that were conducted (e.g., pruning with training).